# Association of Absolute and Relative Handgrip Strength with Prevalent Metabolic Syndrome in Adults: Korea National Health and Nutrition Examination Survey 2014–2018

**DOI:** 10.3390/ijerph191912585

**Published:** 2022-10-02

**Authors:** Sunghyun Hong, Minsuk Oh, Youngwon Kim, Justin Y. Jeon

**Affiliations:** 1Exercise Medicine and Rehabilitation Laboratory, Department of Sport Industry Studies, Yonsei University, Seoul 03722, Korea; 2Department of Public Health, Baylor University, Waco, TX 76798, USA; 3Li Ka Shing Faculty of Medicine, School of Public Health, The University of Hong Kong, Pok Fu Lam, Hong Kong; 4Exercise Medicine Center for Diabetes and Cancer Patients, ICONS, Yonsei University, Seoul 03722, Korea; 5Cancer Prevention Center, Severance Hospital, Yonsei University College of Medicine, Seoul 03722, Korea

**Keywords:** handgrip strength, muscular strength, metabolic syndrome, aging, body mass index

## Abstract

Maintaining or improving muscular strength may be a key preventive strategy for metabolic syndrome (MetS). However, whether the association of handgrip strength (HGS), as a well-established marker of whole-body muscular strength, with the prevalent metabolic syndrome (MetS) varies with age stratification remains unclear. Additionally, whether absolute of relative HGS is superior to another in predicting MetS is less clear. We examined the association of both relative and absolute HGS with the prevalence of MetS in different age groups. Korean adults aged ≥19 years (n = 28,146; 55.7% female) from the Korean National Health and Nutrition Examination Survey (2014–2018) were examined. HGS was categorized using tertile split (highest, intermediate, lowest) and participants were stratified into different age groups at 10-year intervals. Multivariable logistic regression models were used to examine the association between absolute/relative HGS tertiles and MetS with adjustment for covariates. Lower odds of MetS were observed across lower absolute HGS tertiles and the associations were significant in young participants (19–29 years) in both sexes (odds ratio (OR): 0.59 (95% CI: 0.38–0.92) for intermediate and OR: 0.55 (95% CI: 0.34–0.89) for lowest in males; OR: 0.36 (95% CI: 0.20–0.65) for intermediate and OR: 0.42 (95% CI: 0.24–0.74) for lowest in females; all *p* < 0.05). By contrast, higher odds of MetS were observed across lower relative HGS in all age groups in both sexes (in total participants, OR: 2.32 (95% CI: 2.06–2.62) for intermediate and OR: 3.69 (95% CI: 3.27–4.16) for lowest in males and OR: 2.04 (95% CI: 1.83–2.28) for intermediate and OR: 3.28 (95% CI: 2.94–3.65) for lowest in females all *p* < 0.05). The associations of both absolute and relative HGS with MetS attenuated with an increase in age. Our findings suggest that poor relative HGS, as a marker of muscular strength, and not absolute HGS, may be associated with a higher risk of MetS in adults. Our findings also suggest that relative HGS may overestimate MetS in young adults.

## 1. Introduction

Metabolic syndrome (MetS) is a major public health concern due to the pandemic of obesity and sedentary lifestyles [1]. MetS is a complex health disorder that is associated with an increased risk of major chronic diseases, such as diabetes, cardiovascular disease (CVD), as well as some types of cancer [2,3]. MetS is determined when the presence of three or more of the following components is observed: 1) high waist circumference, ≥90 cm in males and ≥80 cm in females (Asia-specific); (2) high SBP, ≥130 mmHg, or high DBP, ≥85 mmHg; (3) high FG level, ≥100 mg/dL; (4) low HDL-C level, <40 mg/dL in males and <50 mg/dL in females; and (5) high TG level, ≥150 mg/dL [4,5]. Hence, a lot of effort in medical research has been devoted to understanding the associations of different conditions with MetS. Substantial evidence suggests that muscular strength is associated with a reduced risk of metabolic disorders [6,7,8,9]. Although the mechanism behind the association between muscular strength and risk of metabolic disorders requires further investigation, maintaining or improving muscular strength is important for the prevention and management of metabolic disorders [6,7,9,10].

Growing evidence suggests that both muscular strength and muscle mass are important parameters for determining health outcomes [11,12,13]. With age, muscle strength declines faster than muscle mass [14]. Moreover, muscle strength can decrease even when muscle mass is maintained or increased [15]. Although there are no established criterion values of Handgrip strength (HGS) for studying health outcomes, HGS is a well-documented predictor for muscular strength [16] as well as the onset of age-related adverse health conditions (e.g., functional limitation and frailty [17], diabetes [10], mortality [18], or cognitive impairment [19]) and MetS [20,21,22,23,24]. HGS has been widely used for estimating muscular strength in population-based clinical and epidemiologic settings as it is easy to measure and relatively cost-effective.

Apart from sex, age, and nutritional status, body weight is also an important determinant for HGS [25]. Therefore, previous studies [21,23,26,27,28] examining the relationship between HGS and MetS have used relative HGS (RHGS), that is, HGS divided by body weight or BMI. Absolute HGS (AHGS) seems to be more closely associated with muscular strength, whereas RHGS seems to be more closely associated with adverse health outcomes (such as MetS, CVD, and all-cause mortality) [9,29,30,31]. Interestingly, Ho et al. [29] recently reported that AHGS and RHGS did not differ in predicting all-cause mortality. However, it is unclear whether AGHS or RHGS is superior over the other in predicting MetS [27,28]. Byeon et al. [27] and Chun et al. [28] reported that low RHGS was associated with increased prevalence of MetS, whereas AHGS was not associated with MetS. These studies were however limited by a relatively small sample size [27] and a sectional population (only older adults) [28], respectively.

Additionally, age should be considered when examining the association of HGS with MetS because body mass increases with age, whereas HGS decreases and the prevalence of MetS also increases. To the best of our knowledge, although previous studies have adjusted for age in the association of HGS with MetS, they have not stratified the findings by age groups, possibly due to the limited sample sizes [21,23,26,32]. Therefore, further examination of the association of both AHGS and RHGS with MetS stratified by age is warranted to understand which expression of HGS is optimal in predicting MetS.

The purpose of this study was to examine the association of AHGS and RHGS with the prevalence of MetS and to compare the pattern and/or magnitude of the association of AHGS vs. RHGS with MetS across different age groups in Korean adults surveyed in the Korea National Health and Nutrition Examination Survey (KNHANES). According to a recent report from the KNHANES, the prevalence of MetS in 2017 is 28.1% and 18.7% in Korean men and women, respectively [33]. KNHANES provided an opportunity to examine the association between HGS and MetS in large, healthy Korean adults with a varied age range. We hypothesized that (1) high AHGS and RHGS are associated with reduced prevalence of MetS and (2) association of AHGS and RHGS with MetS varies across age groups.

## 2. Materials and Methods

### 2.1. Study Participants

We used the data from the 6th (2014–2015) and 7th (2016–2018) KNHANES, an ongoing nationally representative and cross-sectional (retrospective) surveillance system conducted every year by the Korea Centers for Disease Control and Prevention (KCDC) [34]. The datasets used from the current study are available in the KNHANES repository (https://www.data.go.kr/data/15076556/fileData.do (accessed on 1 February 2021)). KNHANES included three component surveys: health interview (face-to-face interview in the mobile examination center; housing characteristics, medical conditions, socioeconomic status, health care utilization, quality of life, injury, lifestyle behaviors, such as smoking, alcohol use, physical activity, oral health, weight control, safety, reproductive health for women), health examination (self-administered in the mobile examination center; body measurements, blood pressure, laboratory test, dental measurement, vision, retinal photo and visual field, audiometry, spirometry, balance, bone density and body composition, chest, knee and hip-joint X-ray), and nutrition survey (face-to-face interview in sample person’s home; dietary behavior, dietary supplement use, food security, food frequency, food and dietary intake). Health interviews and health examinations were conducted by trained medical staff and interviewers using calibrated equipment according to a standardized protocol. The health examinations including anthropometric measurements, laboratory tests (blood profile), and physical examination were performed in a mobile examination center. KNHANES was conducted following ethical approval (No: 2008-04EXP-01-C, 2009-01CON-03-2C, 2010-02CON-21-C, 2011-02CON-06C) by the KCDC Institutional Review Board for the collection of data. The KNHANES data used in the current study are publicly available and an additional ethical approval for data use was not required. Written informed consent was obtained from all participants. Of the 39,199 individuals aged 19 years or older surveyed between 2014 and 2018, we excluded participants who did not have health examination data, such as weight, height, waist circumference, HGS, fasting plasma glucose (FG), HDL-C, triglyceride (TG), and systolic (SBP) and diastolic blood pressure (DBP). Thus, a total of 28,146 participants (12,470 males and 15,676 females) were included in the final analyses (Figure 1). Additional information about the KNHANES data is available elsewhere [34].

### 2.2. Anthropometric and Biochemistry Measures

Height (to the nearest 0.1 cm) and body weight (to the nearest 0.1 kg) were measured using a height-measuring device (Seca 225; GmbH&Co. KG, Hamburg, Germany) and a portable digital scale (GL-6000-20; Caskorea, Seoul, South Korea), respectively. BMI was calculated as weight (kg) divided by height squared (m^2^). Waist circumference (cm) was measured at the point between the lowest rib and the top of the iliac crest using an ergonomic circumference measuring tape (Seca 201; GmbH&Co.KG). Blood pressure was measured on the right arm with the participant in a seated position after a 10 min rest period using a standard mercury sphygmomanometer (Baumanometer, Copiague, NY, USA). Three measurements were recorded at 5 min intervals and an average of the last two measurements was used for analysis. Blood samples were collected from the participants in the morning after an overnight fast for at least 8 h and analyzed at a certified, central laboratory. FG, HDL-C, and TG levels were analyzed using an ADVIA1650 autoanalyzer (Siemens Medical Solutions Diagnostics, Erlangen, Germany). The laboratory performance was monitored by a laboratory data quality control program to ensure that the data met the required standard of accuracy [34].

### 2.3. HGS Measurement

HGS was measured in both the right and left hand using a digital handgrip dynamometer (TKK 5401; Takei, Tokyo, Japan) [22,26]. Participants who had ectrodactyly, fractured fingers, hand paralysis, or any physical problems in handgrip were excluded from the study. The dynamometer was adjusted such that the participants could hold the handle comfortably with their intermediate phalanges flexed at a 90° angle. Participants were asked to extend the arm fully without the hand touching the body. The HGS tests were performed three times alternatively with each hand, with at least 30 s of rest between each trial. The summation of the maximal record from each hand was calculated as AHGS (kg), and RHGS (kg/kg) was calculated as AHGS divided by body weight (kg). Both AHGS and RHGS were categorized using tertile splits (tertile 1 or referent: highest, tertile 2: intermediate, tertile 3: lowest) within the age stratifications. Tertiles of AHGS were 43 kg < highest, 37 kg ≤ intermediate ≤ 43 kg, and lowest < 37 kg in males and 26 kg < highest, 22 kg ≤ intermediate ≤ 26 kg, and lowest < 22 kg in females. Tertiles of RHGS were 0.62 < highest, 0.53 ≤ intermediate ≤ 0.62, and lowest < 0.53 in males and 0.46 < highest, 0.38 ≤ intermediate ≤ 0.46, and lowest < 0.38 in females.

### 2.4. MetS Diagnosis and Covariates

MetS was defined, based on the criteria published by the International Diabetes Foundation [4] and a Joint Interim Statement of the American Heart Association/National Heart, Lung, and Blood Institute (AHA/NHLBI) [5], as the presence of three or more of the following components: (1) high waist circumference, ≥90 cm in males and ≥80 cm in females; (2) high SBP, ≥130 mmHg, or high DBP, ≥85 mmHg; (3) high FG level, ≥100 mg/dL; (4) low HDL-C level, <40 mg/dL in males and <50 mg/dL in females; and (5) high TG level, ≥150 mg/dL.

Sociodemographic factors (including age, sex, education, and income) and lifestyle-related factors (including smoking status, alcohol consumption, and resistance training participation rate) were collected via self-administration using a health interview questionnaire [34]. The level of education (highest degree and years of education) was classified into the following categories: elementary school (≤6 years), middle school (7–9 years), high school (10–12 years), and college or higher (≥13 years). Gross household income was categorized using quartiles as low, mid-low, mid-high, and high. Lifestyle-related factors were categorized as follows: smoking status, current, never or past; frequency of alcohol consumption, none, ≤1 time/month, ≥2–4 times/month, 2–3 times/week, or ≥4 times/week; and frequency of resistance training participation, none, 1–2 days/week, 3–4 days/week, or ≥5 days/week.

### 2.5. Statistical Analyses

Sex-specific descriptive statistics, including frequency distributions and measures of central tendency and variability, were used to calculate and indicate all variables of interest stratified using AHGS tertiles. The differences in variables across HGS tertiles were examined using one-way ANOVA or Kruskal–Wallis tests for continuous variables [35], and chi-square tests for categorical variables [36], as appropriate.

Odds ratios (OR) with 95% confidence intervals were calculated using sex-specific multivariable logistic regression analyses [37] to examine the association of tertiles of both AHGS and RHGS with the prevalence of MetS. Study participants were stratified into six age groups with 10-year intervals: (1) 19–29 years; (2) 30–39 years; (3) 40–49 years; (4) 50–59 years; (5) 60–69 years; and (6) 70–80 years. Results of multivariable logistic regression models with adjustment for covariates (including age, alcohol consumption, smoking status, education, income, and resistance training participation) were also stratified by age groups. All covariates as potential confounders that may affect the association between HGS and MetS were included a priori based on the literature. In sensitivity analyses, we examined RHGS, normalized by BMI, to account for the differences in the association between RHGS and MetS by body mass. A type 1 error level set at <0.05 was considered statistically significant and SPSS version 24.0 (IBM Corp., Armonk, NY, USA) was used for all analyses.

## 3. Results

### 3.1. Baseline Characteristics

The characteristics of participants by AHGS tertile, total and stratified by sex, are shown in Table 1. All participant characteristics were significantly different across AHGS tertiles in both males and females (*p* < 0.01). The male participants were younger, more likely to be obese (higher BMI and waist circumference), with lower SBP, higher DBP, lower FG level, higher HDL-C level, higher TG level, and higher RHGS across higher tertiles of AHGS (*p* < 0.05). The characteristics of female participants across higher tertiles of AHGS were similar to those of male participants, except for BMI, waist circumference, and TG levels, but all characteristics were significantly different across the tertiles (*p* < 0.05). The prevalence of MetS in the first, second, and third tertile of AHGS was 30.5%, 29.6%, and 28.5%, respectively, in males, and 22.6%, 24.4%, and 35.8%, respectively, in females.

### 3.2. Characteristics of MetS Components across AHGS and RHGS Tertiles by Age Groups

The sex-specific characteristics of MetS components across AHGS and RHGS tertiles by age groups are presented in Figure 2 and Figure 3. FG level was higher across lower AHGS among older male participants, and FG level was significantly different across the tertiles of AHGS in the 50’s and 60’s age group (*p* < 0.05). By contrast, significantly higher FG was observed among the female participants with the highest (tertile 1) absolute HGS in the 20’s and 30’s age group. In males, DBP was higher across lower RHGS in the 20–50’s age group, however, this pattern reversed in the 60’s and 70’s age group.

### 3.3. Association of Absolute and Relative HGS with MetS in Males

The association of AHGS and RHGS with MetS in males was examined using multivariable logistic regression models (Table 2). In the AHGS model, compared to the first tertile, there were lower odds of MetS in the second and the third tertiles in both the unadjusted and adjusted models across all age groups (*p* < 0.05), except for the 60–69 years age group. The second and the third tertiles had 17% (OR, 0.83; 95% CI, 0.75–0.93) and 41% lower odds (OR, 0.59; 95% CI, 0.52–0.67) of MetS, respectively, compared to the first tertile in the adjusted model. Conversely, in the relative HGS model, compared to the first tertile, there were higher odds of MetS in the second and the third tertiles in both the unadjusted and adjusted models across all age groups (*p* < 0.05). The second and the third tertiles had 132% (OR, 2.30; 95% CI, 2.07–2.56) and 259% higher odds (OR, 3.59; 95% CI, 2.23–3.99) of MetS, respectively, compared to the first tertile in the adjusted model.

### 3.4. Association of Absolute and Relative HGS with MetS in Females

The association of AHGS and RHGS with MetS in females is presented in Table 3. In the AHGS model, compared to those in the first tertile, there were lower odds of MetS in the second and the third tertile in both the unadjusted and adjusted models in the 19–29, 30–39, and 40–49 years age groups (*p* < 0.05). The second and the third tertiles had 22% (OR, 0.78; 95% CI, 0.70–0.85) and 28% lower odds (OR, 0.72; 95% CI, 0.65–0.79) of MetS, respectively, compared to the first tertile in the adjusted model. Conversely, in the RHGS model, compared to the first tertile, there were higher odds of MetS in the second and third tertiles in both the unadjusted and adjusted models across all age groups (all *p* < 0.05). The second and the third tertiles had 104% (OR, 2.04; 95% CI, 1.83–2.28) and 228% higher odds (OR, 3.28; 95% CI, 2.94–3.65) of MetS, respectively, compared to the first tertile in the adjusted model.

## 4. Discussion

In this large representative sample of Korean adults surveyed in KNHANES, we found that lower AHGS was associated with a lower prevalence of MetS, whereas lower RHGS was associated with a higher prevalence of MetS, independent of all tested covariates. Contrary to our hypothesis, higher AHGS was associated with higher odds of MetS, possibly due to higher waist circumference and/or BMI in individuals with higher AHGS. Furthermore, our findings stratified by age suggest that lower RHGS was associated with higher odds of MetS across all age groups with attenuation of the association with increasing age, whereas the positive association between AHGS and MetS was mostly found in the younger age group.

Consistent with previous studies, RHGS was inversely associated with the prevalence of MetS in older participants [9,21,28], or all age-adjusted adults [26,27] when adjusting for age. Interestingly, the magnitude of association of HGS with MetS was attenuated with increasing age. The mechanism behind the different magnitude of associations by age is unclear; however, we assume that different distribution of impaired MetS components across the age groups may affect the results. Similar to our assumption, a recent report from Teixeira et al. [38] demonstrated that an increase in age may indirectly influence on a higher prevalence of MetS in both sexes. For example, as shown in Appendix A, elevated TG levels in males and lowered HDL-C levels in females in the age groups 19–29 and 30–39 years accounted for the highest proportions of MetS, whereas elevated FG levels in males and elevated waist circumference in females accounted for the highest proportions of MetS in the age groups 50–59, 60–69, and 70–80 years. Although further investigation is warranted to understand whether different distribution of the MetS components modifies the magnitude of association between HGS and MetS in young adults and middle-aged-to-older adults, based on our findings, we suggest that the association between RHGS and MetS may be distinct by age, due to age-related varied distribution of the MetS components. Moreover, greater variability of HGS data and much lower prevalence of MetS in younger participants than in older participants (our preliminary results of data) may also result in the different magnitude of the association of HGS with MetS.

Additionally, our age-stratified results showed that participants with lower RHGS had unfavorable MetS components (except for SBP) across all age groups. Similarly, Sayer et al. [9] also found that lower HGS was associated with high TG levels, waist circumference, and 2 h glucose when adjusting for age and body weight in UK adults aged 59–73 years. Furthermore, Yi et al. [26] reported that increasing quartiles of R HGS were associated with lower odds of having impairment in all MetS components when adjusting for age in Korean adults surveyed in KNHANES. However, whether poor RHGS is associated with adverse MetS components in certain age groups, especially in young adults, remains unclear. Lower muscle mass in older adults is associated with a higher prevalence of MetS in both Asians and Caucasian [39,40]; Kim et al. [40] reported that low muscle mass was associated with a high prevalence of MetS in non-obese young Korean adults aged 19–39 years surveyed in KNHANES. We observed that the pattern of association of RHGS with MetS components is consistent across all age groups. Although MetS components change unfavorably and HGS concurrently decreases with age, we assume that lower RHGS may be associated with MetS in all age groups.

Only a few studies [27,28,32] have examined the association of AHGS and RHGS with MetS in a homogeneous population, and the findings of these studies were consistent with our results. Recently, Byeon et al. [27] and Chun et al. [28] reported that there were no associations between AHGS and MetS in both unadjusted and adjusted regression models in Korean adults surveyed in KNHANES. Although these associations were not statistically significant [27,28], similar to our findings, there was a decreasing trend in the odds of MetS across lower AHGS. We assume that the primary reason for this is because body weight or BMI, the decisive determinant for calculating HGS [25], was not normalized. This further implies that higher body weight and subsequent impairment of MetS components may substantially impact the association of AHGS with a higher prevalence of MetS. In particular, we identified that FG was elevated primarily in older groups across lower AHGS, consistent with the results of previous reports [27,32]. It is well-documented that reductions in muscle mass and subsequent reduction in HGS with age results in poor glucose disposal and affects glucose and muscle metabolism [41]. Additionally, hyperglycemia may directly cause deterioration of muscle contraction and force generation [42]. Therefore, we suggest that there is a bi-directional association between low HGS and high FG with age.

Contrary to our findings, Ho et al. [29] recently reported that AHGS along with RHGS can predict all-cause, CVD and cancer mortality in UK adults. Nevertheless, recent reports by Ho et al. [29] and McGrath et al. [33] suggest that HGS be normalized by BMI, age, and/or sex to examine the association between HGS and adverse health outcomes in different populations. Therefore, based on our findings and those of recent reports [29,33], we suggest that sex, age, and body weight or BMI may be key modifiers in the association between HGS and MetS.

Our study has certain clinical implications. Despite the potential importance of HGS in predicting MetS, as well as other adverse health outcomes, physical function or HGS is not widely measured in the clinical setting (e.g., health care services and hospitals). Therefore, future cardiometabolic disease prevention and/or screening studies, health practices, and medical services should consider HGS, as a simple and safe screening tool. Our findings could be useful to establish health care policies to disseminate the importance of maintaining muscular strength and the advantage of the application of HGS measures, as a proxy measure of total body muscular strength, to the general Korean adulthood population. Furthermore, MetS prevention and intervention strategies should highlight the importance of maintaining or increasing HGS for protection against cardiometabolic diseases in adults of all ages.

The strengths of this study include its large, nationally representative sample of Korean adults, age stratification of results, and uniform collection of biochemistry markers data that is highly reproducible [34]. However, this study has several limitations. First, we cannot infer causality between HGS and MetS due to the cross-sectional observational retrospective study design. Additional research using prospective cohort design or interventional design to examine the temporal association between HGS and incident MetS is warranted. Second, the study findings cannot be generalized to wider populations given that we studied Korean adults surveyed in KNHANES, as there are known racial/ethnic differences in muscular strength as well as risk factors for metabolic diseases. We observed consistent findings with some previous studies with Caucasian participants [9]; however, findings should be interpreted with caution. Third, other muscular strength measures and its associations with MetS may increase the generalizability of the association between HGS and MetS. Future research examining the longitudinal association between HGS and MetS in a study with racially/ethnically diverse participants is needed. Lastly, some potential confounders (such as menstrual cycle, use of certain medications, family history of major metabolic disorders) that might impact the association between HGS and MetS [43,44,45] were not included in the study.

## 5. Conclusions

The association between HGS and MetS was distinct in different age groups. Higher AHGS was associated with a higher prevalence of MetS in younger age groups, whereas lower RHGS was associated with a higher prevalence of MetS across all age groups. These associations were evident after adjusting for all tested covariates. In addition, the association between RHGS and MetS gradually attenuated with age, and therefore, we suggest that RHGS may overestimate the prevalence of MetS in young adults compared to that in older adults. Our findings will help health/medical professionals to utilize HGS as a useful clinical screening tool for metabolic disease risk factors and MetS in adults.

## Figures and Tables

**Figure 1 ijerph-19-12585-f001:**
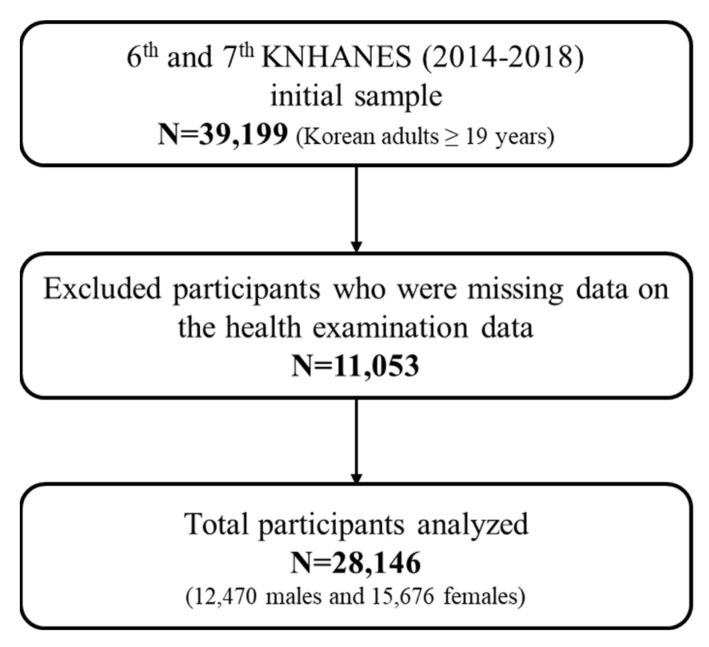
Flow diagram of the analytical participant selection, KNHANES.

**Figure 2 ijerph-19-12585-f002:**
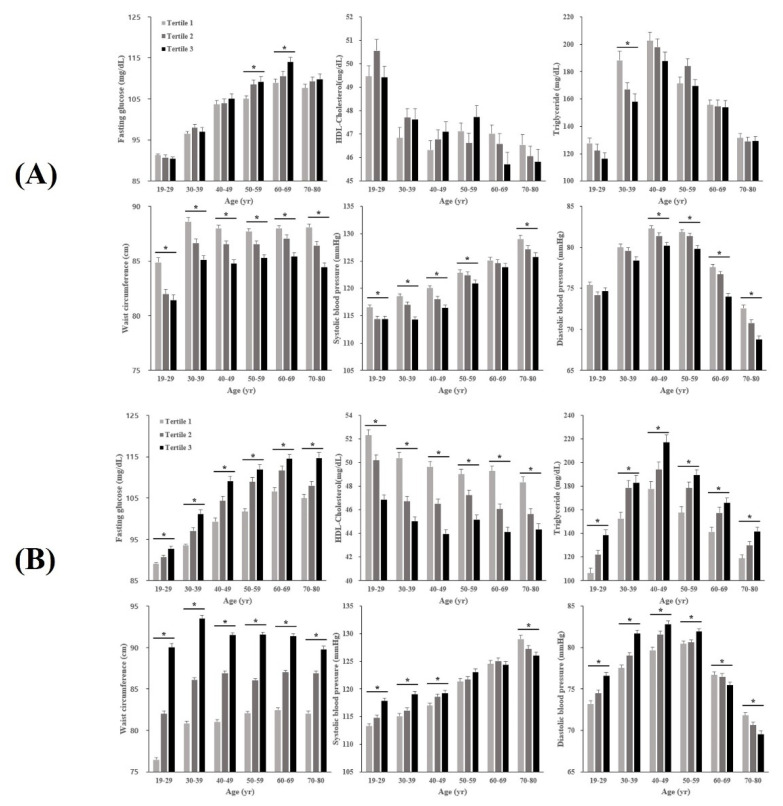
Metabolic syndrome components across absolute (**A**) and relative (**B**) handgrip strength tertiles in men, stratified by age. Note: T1 (first tertile), highest; T2 (second tertile), intermediate; T3 (third tertile), lowest. * *p*-value tests for a difference between groups by handgrip strength tertile using ANOVA (*p* < 0.05). Abbreviations: HDL-C, high-density lipoprotein-cholesterol.

**Figure 3 ijerph-19-12585-f003:**
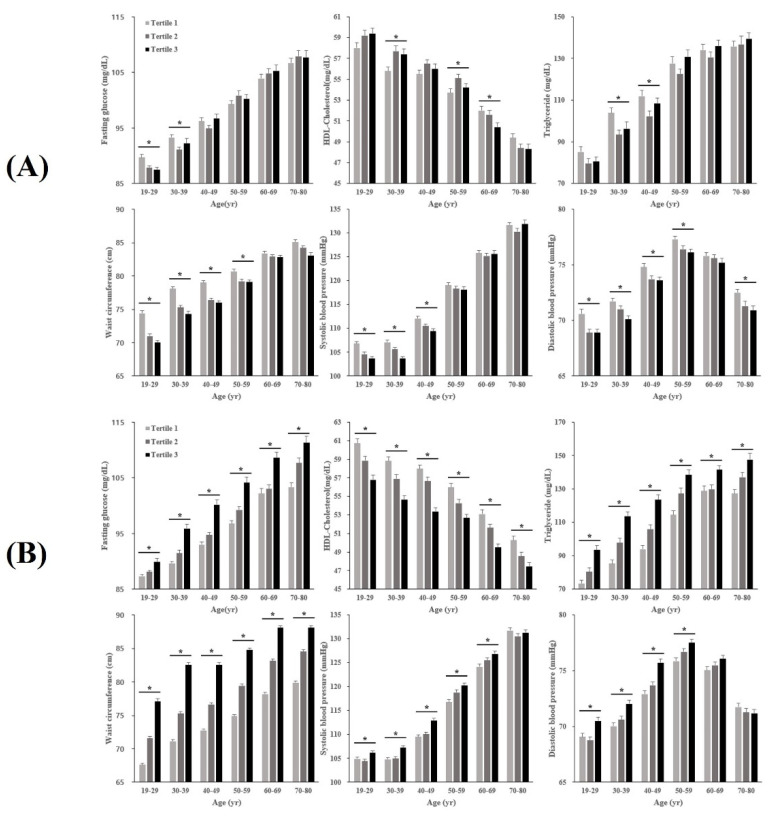
Metabolic syndrome components across absolute (**A**) and relative (**B**) handgrip strength tertiles in women, stratified by age. Note: Note: T1 (first tertile), highest; T2 (second tertile), intermediate; T3 (third tertile), lowest. * *p*-value tests for a difference between groups by handgrip strength tertile using ANOVA (*p* < 0.05). Abbreviations: HDL-C, high-density lipoprotein-cholesterol.

**Table 1 ijerph-19-12585-t001:** Characteristics of participants by tertile of absolute handgrip strength, stratified by sex.

	Males (n = 12,470)				Females (n = 15,676)			
Characteristics	T1 > 43 kg	37 kg ≤ T2 ≤ 43 kg	37 kg > T3	*p*	T1 > 26 kg	22 kg ≤ T2 ≤ 26 kg	22 kg > T3	*p*
	(n = 4379)	(n = 4250)	(n = 3841)		(n = 6327)	(n = 4649)	(n = 4700)	
Age, years	43.1 ± 12.6	50.3 ± 15.6 *	61.4 ± 17.1 **	<0.001	45.4 ± 13.5	50.3 ± 16.0 *	60.0 ± 17.8 **	<0.001
Height, cm	173.7 ± 5.8	170.2 ± 6.0 *	166.2 ± 6.7 **	<0.001	159.9 ± 5.7	157.1 ± 5.9 *	153.4 ± 6.7 **	<0.001
Weight, kg	76.3 ± 11.4	70.6 ± 10.4 *	64.8 ± 10.4 **	<0.001	60.7 ± 9.7	57.1 ± 8.4 *	55.2 ± 8.8 **	<0.001
BMI, kg/m^2^	25.3 ± 3.3	24.4 ± 3.2 *	23.4 ± 3.2 **	<0.001	23.8 ± 3.7	23.2 ± 3.5 *	23.5 ± 3.6 **	<0.001
MetS component								
WC, cm	87.3 ± 8.8	86.1 ± 8.8 *	84.9 ± 9.3 **	<0.001	79.1 ± 9.8	78.3 ± 9.7 *	80.2 ± 10.2 **	<0.001
SBP, mmHg	119.9 ± 13.4	120.8 ± 15.2 *	122.8 ± 16.4 **	<0.001	114.2 ± 16.0	116.0 ± 17.4 *	121.4 ± 19.5 **	<0.001
DBP, mmHg	79.9 ± 9.9	78.1 ± 9.9 *	73.9 ± 10.5 **	<0.001	74.0 ± 9.6	73.2 ± 9.5 *	72.5 ± 9.8 **	<0.001
Glucose, mg/dL	100.8 ± 21.0	104.4 ± 27.4 *	107.4 ± 29.4 **	<0.001	96.6 ± 18.6	98.0 ± 23.1 *	102.0 ± 26.0 **	<0.001
HDL-C, mg/dL	47.3 ± 10.9	47.5 ± 11.3	46.6 ± 11.8 **	0.002	55.1 ± 12.6	55.0 ± 12.8	52.2 ± 12.5 **	<0.001
TG, mg/dL	172.7 ± 144.8	162.7 ± 136.7 *	144.5 ± 107.3 **	<0.001	112.8 ± 86.5	112.3 ± 74.3	124.7 ± 84.6 **	<0.001
Absolute handgrip strength, kg	48.8 ± 4.4	40.0 ± 2.0 *	30.9 ± 4.8 **	<0.001	29.1 ± 2.8	23.5 ± 1.1 *	17.9 ± 2.9 **	<0.001
Relative handgrip strength	0.65 ± 0.1	0.58 ± 0.1 *	0.49 ± 0.1 **	<0.001	0.49 ± 0.1	0.42 ± 0.1 *	0.33 ± 0.1 **	<0.001
No. of MetS components, n (%) ^a^				<0.001				<0.001
1	1046 (24.6)	1000 (24.4)	928 (26.1)		1251 (25.4)	1388 (25)	901 (21.2)	
2	986 (23.2)	988 (24.1)	921 (26)		1025 (20.8)	1042 (18.8)	916 (21.6)	
3	726 (17.1)	736 (18)	614 (17.3)		652 (13.2)	792 (14.3)	851 (20.1)	
4	437 (10.3)	384 (9.4)	328 (9.2)		370 (7.5)	437 (7.9)	501 (11.8)	
5	134 (3.2)	92 (2.2)	71 (2)		92 (1.9)	125 (2.3)	168 (4)	
MetS	1297 (30.5)	1212 (29.6)	1013 (28.5)	0.004	1114 (22.6)	1354 (24.4)	1520 (35.8)	<0.001
Alcohol consumption				<0.001				<0.001
None	318 (7.3)	487 (11.5)	741 (19.3)		1070 (16.9)	827 (17.8)	996 (21.2)	
≤1 time/month	871 (19.9)	888 (20.9)	718 (18.7)		2261 (35.8)	1608 (34.6)	1343 (28.6)	
2–4 time/month	1308 (29.9)	1112 (26.2)	774 (20.2)		1420 (22.5)	913 (19.7)	572 (12.2)	
2–3 times/week	1154 (26.4)	951 (22.4)	665 (17.3)		695 (11.0)	405 (8.7)	270 (5.7)	
≥4 times/week	495 (11.3)	545 (12.8)	520 (13.5)		154 (2.4)	115 (2.5)	111 (2.4)	
Smoking status				<0.001				<0.001
Never or past	2470 (56.4)	2663 (62.7)	2680 (69.8)		5866 (92.8)	4330 (93.2)	4342 (92.4)	
Current	1786 (40.8)	1485 (34.9)	1031 (26.8)		353 (5.6)	223 (4.8)	176 (3.7)	
Education				<0.001				<0.001
Elementary school	123 (2.8)	380 (8.9)	906 (23.6)		627 (9.9)	925 (19.9)	1891 (40.3)	
Middle school	235 (5.4)	449 (10.6)	562 (14.6)		557 (8.8)	538 (11.6)	516 (11.0)	
High school	1204 (27.5)	1150 (27.1)	853 (22.2)		1968 (31.1)	1172 (25.2)	792 (16.9)	
College or higher	2139 (48.8)	1727 (40.6)	1016 (26.5)		2593 (41)	1586 (34.2)	1030 (21.9)	
Income (quartile)				<0.001				<0.001
Lowest	296 (6.8)	589 (13.9)	1238 (32.2)		627 (9.9)	841 (18.1)	1687 (35.9)	
Mid–low	955 (21.8)	1037 (24.4)	1011 (26.3)		557 (8.8)	1171 (25.2)	1104 (23.5)	
Mid–high	1506 (34.4)	1216 (28.6)	799 (20.8)		1968 (31.1)	1298 (28.0)	929 (19.8)	
Highest	1616 (36.9)	1391 (32.7)	763 (19.9)		2593 (41.0)	1317 (28.4)	952 (20.3)	
Resistance training (days/week)				<0.001				<0.001
None	2511 (57.3)	2658 (62.5)	2633 (68.5)		668 (10.6)	3685 (79.3)	3894 (82.9)	
1–2	522 (11.9)	425 (10.0)	244 (6.4)		1556 (24.6)	301 (6.5)	175 (3.7)	
3–4	534 (12.2)	407 (9.6)	255 (6.6)		1974 (31.2)	245 (5.3)	155 (3.3)	
≥5	455 (10.4)	508 (12.0)	420 (10.9)		2096 (33.2)	202 (4.3)	136 (2.9)	

Note: T1 (first tertile), highest; T2 (second tertile), intermediate; T3 (third tertile), lowest. Data are presented as mean ± SD or n (%). ^a^ The number of participants by the number of elevated or lowered levels of MetS components. *p*-value for trend, difference by handgrip strength tertile, was analyzed using ANOVA, Kruskal–Wallis test, or chi-square test, as appropriate. * Significantly different from tertile 1 (*p* < 0.05); ** significantly different from tertile 2 (*p* < 0.05) using Bonferroni post hoc tests. Abbreviations: BMI, body mass index; WC, waist circumference; SBP, systolic blood pressure; DBP, diastolic blood pressure; HDL-C, high-density lipoprotein cholesterol; TG, triglyceride; MetS, metabolic syndrome.

**Table 2 ijerph-19-12585-t002:** Association between tertiles of handgrip strength and the prevalence of metabolic syndrome in males, stratified by age.

Age (Years)	Absolute HGS	n	Unadjusted	Adjusted	Relative HGS	n	Unadjusted	Adjusted
			Odds Ratio (95% CI)				Odds Ratio (95% CI)	
Total	43 kg < T1	4379	Referent	Referent	0.62 < T1	4159	Referent	Referent
	37 kg ≤ T2 ≤ 43 kg	4250	0.94 (0.86–1.04)	**0.83 (0.75–0.93)**	0.53 ≤ T2 ≤ 0.62	4154	**2.30 (2.07–2.56)**	**2.32 (2.06–2.62)**
	37 kg > T3	3841	**0.85 (0.77–0.94)**	**0.59 (0.52–0.67)**	0.53 > T3	4157	**3.59 (2.23–3.99)**	**3.69 (3.27–4.16)**
19–29	44 kg < T1	585	Referent	Referent	0.63 kg < T1	524	Referent	Referent
	39 kg ≤ T2 ≤ 44 kg	513	**0.59 (0.39–0.88)**	**0.59 (0.38–0.92)**	0.54 ≤ T2 ≤ 0.63	523	**5.93 (2.76–12.71)**	**6.03 (2.65–13.70)**
	39 kg > T3	472	**0.57 (0.37–0.87)**	**0.55 (0.34–0.89)**	0.54 > T3	523	**14.13 (6.79–29.42)**	**16.26 (7.37–35.89)**
30–39	47 kg < T1	741	Referent	Referent	0.65 < T1	651	Referent	Referent
	42 kg ≤ T2 ≤ 47 kg	621	**0.59 (0.46–0.76)**	**0.53 (0.40–0.71)**	0.55 ≤ T2 ≤ 0.65	651	**2.36 91.75–3.19)**	**2.43 (1.72–3.42)**
	42 kg > T3	590	**0.61 (0.48–0.79)**	**0.60 (0.45–0.80)**	0.55 > T3	650	**5.29 (3.97–7.04)**	**6.15 (4.41–8.57)**
40–49	46 kg < T1	766	Referent	Referent	0.65 < T1	741	Referent	Referent
	41 kg ≤ T2 ≤ 46 kg	781	0.81 (0.66–1.00)	0.84 (0.66–1.07)	0.55 ≤ T2 ≤ 0.65	740	**2.33 (1.83–2.98)**	**2.50 (1.89–3.31)**
	41 kg > T3	673	**0.69 (0.55–0.86)**	**0.74 (0.57–0.95)**	0.55 > T3	739	**4.39 (3.46–5.57)**	**5.34 (4.05–7.05)**
50–59	43 kg < T1	848	Referent	Referent	0.64 < T1	793	Referent	Referent
	39 kg ≤ T2 ≤ 43 kg	781	0.99 (0.81–1.21)	1.05 (0.84–1.31)	0.55 ≤ T2 ≤ 0.64	791	**1.68 (1.34–2.11)**	**1.76 (1.38–2.26)**
	39 kg > T3	747	**0.78 (0.64–0.97)**	**0.78 (0.61–0.99)**	0.55 > T3	792	**3.70 (2.98–4.61)**	**4.09 (3.20–5.22)**
60–69	39 kg < T1	883	Referent	Referent	0.61 < T1	753	Referent	Referent
	35 kg ≤ T2 ≤ 39 kg	724	0.93 (0.76–1.15)	1.02 (0.80–1.29)	0.52 ≤ T2 ≤ 0.61	749	**2.02 (1.60–2.55)**	**2.16 (1.67–2.80)**
	35 kg > T3	646	0.88 (0.71–1.09)	0.90 (0.70–1.16)	0.52 > T3	751	**3.15 (2.51–3.96)**	**3.55 (2.74–4.59)**
70–80	34 kg < T1	711	Referent	Referent	0.55 < T1	701	Referent	Referent
	29 kg ≤ T2 ≤ 34 kg	742	**0.75 (0.60–0.94)**	**0.75 (0.58–0.97)**	0.46 ≤ T2 ≤ 0.55	698	**1.55 (1.22–1.99)**	**1.60 (1.22–2.11)**
	29 kg > T3	646	**0.58 (0.45–0.73)**	**0.63 (0.47–0.85)**	0.46 > T3	700	**2.10 (1.65–2.67)**	**2.58 (1.95–3.41)**

Note: T1 (first tertile), highest; T2 (second tertile), intermediate; T3 (third tertile), lowest. Adjusted model includes age, alcohol consumption, smoking status, education, income, and resistance training participation. Bold values are statistically significant (*p* < 0.05). Abbreviations: HGS, handgrip strength; CI, confidence interval.

**Table 3 ijerph-19-12585-t003:** Association between tertiles of handgrip strength and the prevalence of metabolic syndrome in females, stratified by age.

Age (Years)	Absolute HGS	n	Unadjusted	Adjusted	Relative HGS	n	Unadjusted	Adjusted
			Odds Ratio (95% CI)				Odds Ratio (95% CI)	
Total	26 kg < T1	6320	Referent	Referent	0.46 < T1	5224	Referent	Referent
	22 kg ≤ T2 ≤ 26 kg	4644	1.08 (0.99–1.18)	**0.78 (0.70–0.85)**	0.38 ≤ T2 ≤ 0.46	5220	**2.50 (2.25–2.78)**	**2.04 (1.83–2.28)**
	22 kg > T3	4697	**1.70 (1.56–1.86)**	**0.72 (0.65–0.79)**	0.38 > T3	5217	**5.37 (4.85–5.94)**	**3.28 (2.94–3.65)**
19–29	26 kg < T1	677	Referent	Referent	0.48 < T1	617	Referent	Referent
	23 kg ≤ T2 ≤ 26 kg	603	**0.34 (0.19–0.62)**	**0.36 (0.20–0.65)**	0.41 ≤ T2 ≤ 0.48	616	**2.72 (1.06–6.99)**	**2.70 (1.05–6.96)**
	23 kg > T3	569	**0.41 (0.23–0.73)**	**0.42 (0.24–0.74)**	0.41 > T3	616	**10.38 (4.44–24.27)**	**10.08 (4.31–23.57)**
30–39	27 kg < T1	1022	Referent	Referent	0.49 < T1	834	Referent	Referent
	24 kg ≤ T2 ≤ 27 kg	807	**0.62 (0.46–0.85)**	**0.63 (0.47–0.86)**	0.42 ≤ T2 ≤ 0.49	832	**2.49 (1.57–3.95)**	**2.55 (1.61–4.05)**
	24 kg > T3	670	**0.56 (0.40–0.78)**	**0.59 (0.42–0.83)**	0.42 > T3	833	**7.11 (4.67–10.82)**	**7.48 (4.90–11.42)**
40–49	27 kg < T1	1116	Referent	Referent	0.49 kg < T1	967	Referent	Referent
	24 kg ≤ T2 ≤ 27 kg	959	**0.52 (0.41–0.66)**	**0.52 (0.41–0.66)**	0.41 ≤ T2 ≤ 0.49	967	**1.91 (1.43–2.55)**	**1.90 91.42–2.54)**
	24 kg > T3	824	**0.66 (0.52–0.84)**	**0.65 (0.51–0.83)**	0.41 > T3	965	**4.05 (3.09–5.30)**	**3.97 (3.03–5.20)**
50–59	26 kg < T1	1130	Referent	Referent	0.47 < T1	1016	Referent	Referent
	23 kg ≤ T2 ≤ 26 kg	1059	0.86 (0.71–1.03)	0.85 (0.70–1.02)	0.39 ≤ T2 ≤ 0.47	1022	**2.24 (1.80–2.79)**	**2.22 (1.78–2.77)**
	23 kg > T3	868	0.91 (0.75–1.11)	0.89 (0.73–1.09)	0.39 > T3	1019	**3.76 (3.03–4.66)**	**3.72 (3.00–4.61)**
60–69	24 kg < T1	1017	Referent	Referent	0.44 < T1	889	Referent	Referent
	21 kg ≤ T2 ≤ 24 kg	918	0.92 (0.76–1.10)	0.89 (0.74–1.07)	0.36 ≤ T2 ≤ 0.44	887	**1.60 (1.32–1.95)**	**1.57 (1.29–1.91)**
	21 kg > T3	728	1.03 (0.85–1.25)	0.97 (0.80–1.18)	0.36 > T3	887	**2.74 (2.25–3.33)**	**2.65 (2.18–3.23)**
70–80	20 kg < T1	1043	Referent	Referent	0.39 < T1	899	Referent	Referent
	17 kg ≤ T2 ≤ 20 kg	862	0.88 (0.73–1.05)	0.90 (0.75–1.08)	0.31 ≤ T2 ≤ 0.39	898	**1.48 (1.22–1.78)**	**1.52 (1.26–1.84)**
	17 kg > T3	789	0.87 (0.72–1.04)	0.90 (0.74–1.10)	0.31 > T3	897	**1.90 (1.58–2.29)**	**2.02 (1.66–2.45)**

Note: T1 (first tertile), highest; T2 (second tertile), intermediate; T3 (third tertile), lowest. Adjusted model includes age, alcohol consumption, smoking status, education, income, and resistance training participation. Bold values are statistically significant (*p* < 0.05). Abbreviations: HGS, handgrip strength; CI, confidence interval.

## Data Availability

The datasets used from the current study are available in the KNHANES repository (https://www.data.go.kr/data/15076556/fileData.do (accessed on 1 February 2021)).

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
