# Peer review of "Association of Absolute and Relative Handgrip Strength with Prevalent Metabolic Syndrome in Adults: Korea National Health and Nutrition Examination Survey 2014–2018"

_ijerph, 2022, doi:10.3390/ijerph191912585_

Round 1
Reviewer 1 Report
Dear authors,
This study examined the association of both relative and absolute HGS with the prevalence of MetS in different age groups in Korean adults.
The manuscript has some interesting points, however the research gap should be improved. Indeed, although the novelty of the article is poor, the article presented a well-described methodology and results.
Thus, the authors should consider the following major revisions in order to make the manuscript valid for publication with IJERPH standards (please, see attachment).
Good work

Author Response
Please see the attachement.

Reviewer 2 Report
This paper is interesting and important for understanding Association of Absolute and Relative Handgrip Strength with Prevalent Metabolic Syndrome. I suggest some revision for improving quality of manuscript.
Introduction Section
(Comment 1) I recommend authors to supplement definition and statistical information of Metabolic syndrome and Handgrip strength
Materials and Methods
(Comment 2) I recommend authors to supplement selection process of study populations as a figure. (line 83-102)
(Comment 3) I recommend authors to supplement reference of previous studies related to HGS. (line 122-131)
Discussion Section
(Comment 4) I recommend authors to supplement policy development point or suggestion for future study based on the result.
Author Response
Please see the attachement.

Reviewer 3 Report
The article is well structured, the sections are carefully worked out, the materials and methods are correctly and fully described. The authors discussed their results well and synthesized them with world data on this problem.
The design of the study is well planned, the data obtained are sufficient for correct reasoning and conclusions. Logical and understandable research methods were chosen and a large representative sample (n= 28,146) was processed. Statistics are well described and applied.
I agree with the authors that with an increase in absolute grip strength, contrary to expectation, the likelihood of metabolic syndrome increased precisely because individuals with high absolute grip strength weighed more and were larger, respectively, had a higher body mass index and waist circumference.
At the same time, when adjusting grip strength for body mass index and using such a marker as relative grip strength, it was shown that when it decreases, the likelihood of metabolic syndrome increases in all age groups.
That being said, it appears to be a reasonable hypothesis that absolute grip strength appears to be more closely associated with muscle strength, while relative grip strength appears to be more closely associated with adverse health outcomes (such as metabolic syndrome, cardiovascular diseases and mortality from all causes). The data obtained by the authors is one of the confirmations of this hypothesis.
The article is well written and understandable. A detailed check of tables and figures shows high quality and a good choice of the form of data presentation, which correspond to the objectives of the study. Throughout the manuscript, the data are correctly and consistently interpreted. I am impressed by the fact that in the process of describing and discussing their results, the authors make competent comments and conclusions, answer all the questions that they set themselves at the beginning of the work. The limitations and advantages of the work are described in detail and correctly.
I support the conclusion that body mass index corrected grip strength should be considered as a simple and safe screening tool in large studies.
The results of their own research are quite competently compared with the literature on this topic, 54.5% of the sources of the last decade, more than 32.4% of the last 5 years. 40.5% of the sources are older than 10 years, which is too much for an article applying for publication in a Q1 journal. In this case, this is an insignificant remark, since the researchers refer to fairly high-quality works, and otherwise the article was made very carefully and competently, even minor technical remarks are practically absent. Minor note: Literature item No. 38 should be removed, it is empty.
GENERAL CONCLUSION: I liked the work. The manuscript can be accepted for publication in a journal, there are practically no comments.
Author Response
We appreciate the thoughtful comments and analytical view by Reviewer 3. Reviewer 3 highlighted the strengths and implications of our manuscript. We thank Reviewer 3 for her/his contributions and time.
Thank you.